# On the emergence of P-Loop NTPase and Rossmann enzymes from a Beta-Alpha-Beta ancestral fragment

Liam M Longo[1†‡], Jagoda Jabłońska[1], Pratik Vyas[1], Manil Kanade[1§], Rachel Kolodny[2]*, Nir Ben-Tal[3]*, Dan S Tawfik[1]*

[1]Weizmann Institute of Science, Department of Biomolecular Sciences, Rehovot, Israel; [2]University of Haifa, Department of Computer Science, Haifa, Israel; [3]Tel Aviv University, George S. Wise Faculty of Life Sciences, Department of Biochemistry and Molecular Biology, Tel Aviv, Israel

**Abstract** This article is dedicated to the memory of Michael G. Rossmann. Dating back to the last universal common ancestor, P-loop NTPases and Rossmanns comprise the most ubiquitous and diverse enzyme lineages. Despite similarities in their overall architecture and phosphate binding motif, a lack of sequence identity and some fundamental structural differences currently designates them as independent emergences. We systematically searched for structure and sequence elements shared by both lineages. We detected homologous segments that span the first βαβ motif of both lineages, including the phosphate binding loop and a conserved aspartate at the tip of β2. The latter ligates the catalytic metal in P-loop NTPases, while in Rossmanns it binds the nucleotide's ribose moiety. Tubulin, a Rossmann GTPase, demonstrates the potential of the β2-Asp to take either one of these two roles. While convergence cannot be completely ruled out, we show that both lineages likely emerged from a common βαβ segment that comprises the core of these enzyme families to this very day.

*For correspondence:
trachel@cs.haifa.ac.il (RK);
bental@tauex.tau.ac.il (NB-T);
dan.tawfik@weizmann.ac.il (DST)

Present address: †Tokyo Institute of Technology, Earth-Life Science Institute, Tokyo, Japan; ‡Blue Marble Space Institute of Science, Seattle, United States; §Sincrotrone Trieste S.C.p.A, Trieste, Italy

Competing interests: The authors declare that no competing interests exist.

## Introduction

In 1970, Michael Rossmann reported the structure of the first αβα sandwich protein, lactate dehydrogenase (*Adams et al., 1970*). This NAD-utilizing enzyme would later become representative of what is now known as the '*Rossmann fold*' (*Rossmann et al., 1974*). About a decade later, on the basis of a sequence analysis, another major αβα sandwich domain that utilizes phosphorylated nucleosides was proposed (*Walker et al., 1982*), which is now known as the P-loop NTPase, or '*P-loop*' for short. The importance of these two evolutionary lineages, Rossmanns and P-loops, cannot be overstated: Both lineages have diversified extensively, and each is individually associated with more than 120 families and 75 different enzymatic reactions (see *Methods*). Furthermore, these two lineages are ubiquitous across the tree of life (*Leipe et al., 2003*). Accordingly, essentially all studies aimed at unraveling the history of protein evolution concluded that these enzymes emerged well before the last universal common ancestor (LUCA), and were among the very first, if not the first, enzyme families (*Leipe et al., 2003*; *Ma et al., 2008*; *Edwards et al., 2013*; *Alva et al., 2015*; *Aravind et al., 2002a*; *Goncearenco and Berezovsky, 2015*). Indeed, both P-loops and Rossmanns are dubbed nucleotide-binding domains because they both make use of phosphorylated ribonucleosides such as ATP or NAD, as well as of other pre-LUCA cofactors such as SAM (*Laurino et al., 2016*).

As elaborated in the next section, the P-loop and Rossmann domains share a number of similar features, but also some distinct differences. Given their pre-LUCA origin, a common P-loop/Rossmann ancestor – even if it did exist at some point – is surely lost to time. Both lineages emerged during the so-called 'big bang' of protein evolution, an event that marks the birth of the major protein

classes, yet occurred too early to be reconstructed by phylogenetic means (*Aravind et al., 2002a*). Thus, a fundamental enigma surrounding the birth of the first enzymes is whether the Rossmann and the P-loop lineages diverged from a common ancestor, or perhaps, given that they both make use of phosphorylated ribonucleosides, have converged to similar structural and functional features. The former is the more evolutionarily appealing scenario, yet the latter is as common and tangible (*Galperin and Koonin, 2012*; *Elias and Tawfik, 2012*).

To address this longstanding question, we performed a detailed analysis looking for indications of common ancestry with respect to the core elements of these two classes, namely their most conserved and functionally critical structural elements. Indeed, global sequence homology between these lineages, or even shared short sequence motifs, cannot be detected. As such, large-scale analyses of protein homology (*Alva et al., 2015*), including SCOPe (*Chandonia et al., 2017*) and the Evolutionary Classifications of Protein Domains (ECOD) database (*Cheng et al., 2014*) classify P-loop NTPases and Rossmanns as independent evolutionary emergences. However, loss of detectable sequence homology would be expected between lineages that split in the distant past, especially if both have diverged extensively, as is the case for P-loops and Rossmanns. Nonetheless, structural anatomy (*Laurino et al., 2016*) and sophisticated ways of detecting sequence homology may assign common ancestry in highly diverged lineages on the basis of a few common sequence-structure features (*Galperin and Koonin, 2012*; *Elias and Tawfik, 2012*; *Hildebrand et al., 2009*; *Nepomnyachiy et al., 2017*). Further, parallel evolution may operate, with relics of an ancient common ancestor surfacing sporadically in contemporary proteins, thus resulting in detectable sequence and/or structural homology. Thus, if P-loops and Rossmanns do share common ancestry, we might expect the existence of 'bridge proteins'; that is, proteins belonging to one lineage with features that are distinct for the other lineage.

Here, we report the detection and analysis of common features and bridge proteins between P-loops and Rossmanns. The existence of such common features and bridge proteins supports common ancestry, though it does not rule out convergence. Nonetheless, our results suggest what the key features of the ancestor(s) might have been, and indicate that even if these lineages emerged independently, their ancestors shared the very same features. To best frame this analysis, however, we must first dissect the canonical features of Rossmann and P-loop proteins.

## Results and discussion

### P-loop and Rossmann – similar but distinctly different

Both P-loops and Rossmanns adopt the αβα 3-layer sandwich fold (*Figure 1*). This fold, which comprises a parallel β-sheet sandwiched between two layers of α-helices, is among the most ancient, if not the most ancient, protein folds known (*Ma et al., 2008*; *Aravind et al., 2002a*; *Bukhari and Caetano-Anollés, 2013*; *Winstanley et al., 2005*). In essence, αβα sandwich proteins consist of a tandem repeat of β-loop-α elements, where the loops form the active-site (hereafter referred to as the 'functional' or 'top' loops; *Figure 1A*). The minimal P-loop or Rossmann domain comprises five β-loop-α elements linked via short 'connecting' or 'bottom' loops that generally have no functional role. Although many domains have six strands, and sometimes more, we will hereafter consider the minimal 5-stranded core domain for simplicity.

While the overall fold is conserved, the topology – specifically, the strand order of the interior β-sheet – differs between Rossmanns and P-loops. The Rossmann topology (β3-β2-β1-β4-β5) has a pseudo-2-fold axis of symmetry between β1-β3 and β4-β5 (*Figure 1B*; or β1-β3 and β4-β6 in the common 6-stranded domains). However, in the P-loop topology, at least two strands are swapped (*Figure 1C*). The most common P-loop topology is β2-β3-β1-β4-β5 (*Leipe et al., 2003*); although, as discussed below, P-loops can adopt several different strand topologies.

The second shared hallmark is that both P-loops and Rossmans bind phosphorylated ribonucleoside ligands as substrates, co-substrates or cofactors (hereafter, phospho-ligands). While the overall mode of ligand binding differs, the binding modes of their phosphate moieties share a few similarities (*Figure 2A,B*): (*i*) The phosphate is bound by the first β-loop-α element which resides in the center of the domain (hereafter, β1-(phosphate binding loop)-α1); (*ii*) both phosphate binding loops mediate binding via a 'nest' of hydrogen bonds formed by backbone amides at the N-terminus of the first canonical α-helix (α1) as well as via residues from the loop itself; and (*iii*) both phosphate

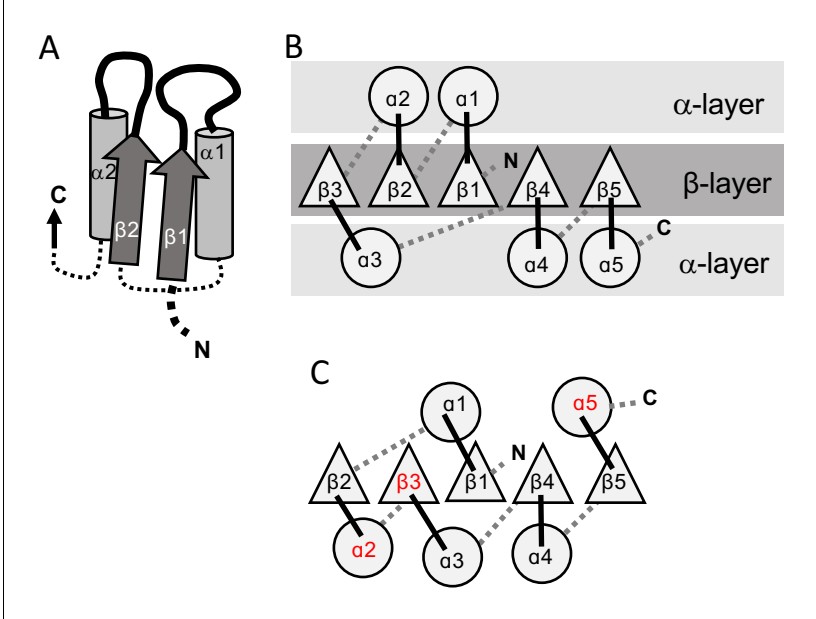

**Figure 1.** The 3-layer αβα sandwich. (**A**) The αβα sandwich is a modular fold comprised of repeating β-loop-α elements. This side view shows two tandem βα elements: the functional loops are situated on the 'top' of the fold (thick lines) and the β-loop-α element are linked via short, bottom loops (thin, dashed lines). Shown here are the first two elements with a Rossmann topology, beginning with β1 at the N-terminus, and the first two helices (α1, α2) that, in this cartoon, comprise one external layer of the sandwich. (**B**) A view from the top reveals the αβα sandwich architecture with its three layers: a parallel β-sheet flanked on both sides by α-helices. The top, active-site loops face the reader whereas the N- and C-termini and the bottom, connecting loops face the back of the page. The order of the β-strands in the interior β-sheet follows the canonical Rossmann topology. (**C**) The most common, core P-loop NTPase (P-loops) topology. Noted in red are the differences from the Rossmann topology—migration of β3 from the edge to the center, and of α2 and α5 from one external layer to another.

binding loops are glycine-rich sequences with similar patterns: the canonical Rossmann motif is GxGxxG, while the canonical P-loop motif, dubbed Walker A, is GxxGxGK[T/S]. To avoid confusion, P-loop is used here to refer to the evolutionary lineage of P-loop NTPases only. When referring to the phosphate binding element of a protein, with no relation to a specific protein lineage, phosphate binding loop (or PBL) is used. Hence, P-loop PBL relates to the phosphate binding loop of P-loop NTPases (the Walker A motif), and Rossmann PBL to the Rossmann's phosphate binding loop. The structural segment in which the phosphate binding loop resides is accordingly dubbed β1-PBL-α1 or, more simply, β-PBL-α.

However, despite similar phosphate binding elements, the mode of phospho-ligand binding by Rossmanns and P-loops is fundamentally different, and this difference relates to important functional differences between the two lineages. Although Rossmann and P-loop proteins both utilize phosphorylated nucleosides, the phosphate groups of these metabolites play a fundamentally different role. P-loops primarily catalyze phosphoryl transfer (including to water, i.e., hydrolysis) and thus most often operate on ATP and GTP with the help of a metal dication. Rossmanns, on the other hand, primarily use NAD(P), with the phosphate moieties serving only as a handle for binding, while the redox chemistry occurs elsewhere (*e.g.*, the nicotinamide base in NAD$^+$). These functional differences are accompanied by a number of structural differences in the mode of phosphate binding: The P-loop Walker A is a relatively long, surface-exposed loop that extends beyond the protein's core and wraps, like the palm of a hand, around the phosphate moieties of the ligand (*Figure 2A*). The Rossmann PBL, however, is short and forms a flat interaction surface, with the phosphate groups interacting mostly with the N-terminus of α1 via a highly conserved and ordered water molecule (*Figure 2B*; *Bottoms et al., 2002*). Foremost, the orientation of the phospho-ligand being bound is different: The nucleoside moiety in Rossmanns is oriented 'inside', that is, in the direction of the β-

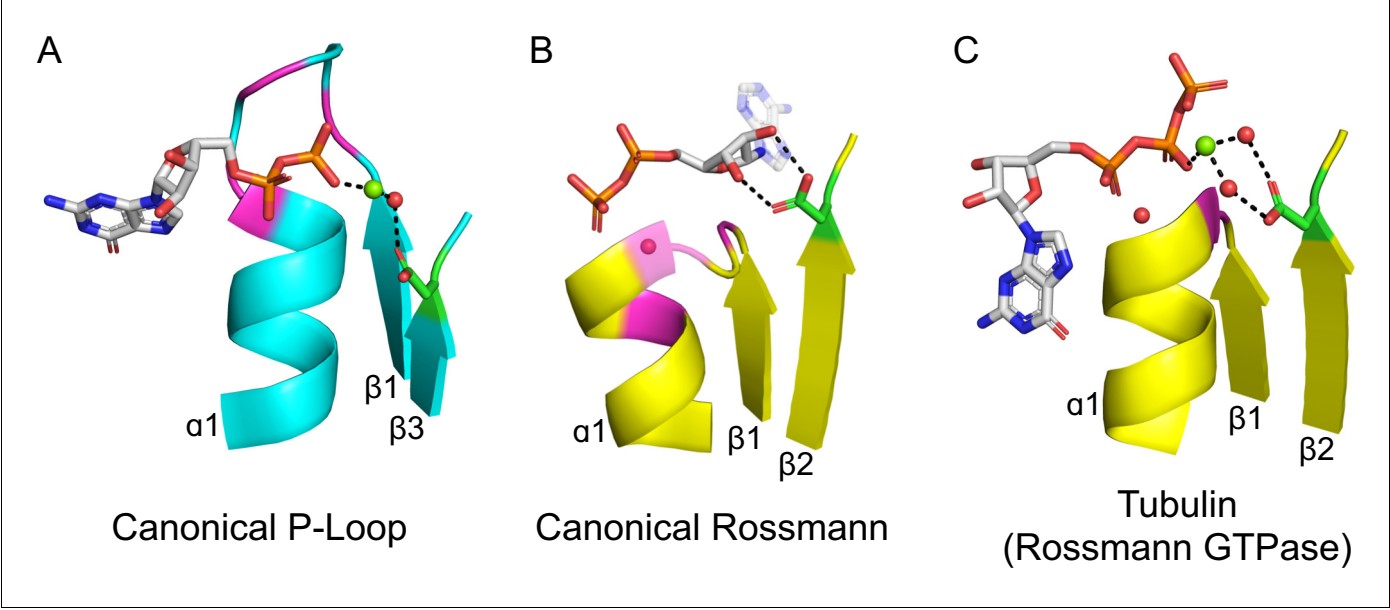

**Figure 2.** The ligand-binding modes of Rossman and P-loop proteins. The phosphate binding loops (PBLs) of both lineages connect the C-terminus of β1 to the N-terminus of α1 (conserved glycine residues are colored magenta). The Rossmann β2-Asp, and the P-loop Walker B-Asp, are in green sticks. Water molecules are denoted by red spheres, and metal dications by green spheres. (**A**) The canonical P-loop NTPase binding mode. The phosphate binding loop (the P-loop Walker A motif; GxxGxGK(T/S)) begins with the first conserved Gly residue at the tip of β1 and ends with a Thr/Ser residing within α1. The Walker B-Asp, located at the tip of β3, interacts with the catalytic $Mg^{2+}$, either directly or via a water molecule, as seen here. (**B**) The canonical Rossmann binding mode. The phosphate binding site includes a canonical water molecule (α1 has been rendered transparent so that the conserved water is visible). The Asp sidechain at the tip of β2 (β2-Asp) forms a bidentate interaction with both hydroxyls of the ribose. Note also the opposite orientations of the ribose and adenine moieties in P-loops (pointing away from the β-sheet) *versus* Rossmann (pointing towards the β-sheet). (**C**) Tubulin is a GTPase that belongs to the Rossmann lineage. It possesses the canonical Rossmann strand topology, phosphate binding loop (including the mediating water), and β2-Asp. However, the ligand, GTP, is bound in the P-loop NTPase mode (as in **A**). Accordingly, the β2-Asp makes a water mediated interaction with the catalytic metal cation ($Ca^{2+}$or $Mg^{2+}$) thus acting in effect as a Walker B-Asp (the metal cation's coordination schemes are also identical, see *Figure 2—figure supplement 3*). ECOD domains used in this figure, from left to right, are e1yrbA1, e1lssA1, and e5j2tB1. All structure figures were prepared in PyMOL (pymol.org).

The online version of this article includes the following figure supplement(s) for figure 2:

**Figure supplement 1.** Rossmann domain binding ATP in the canonical binding mode.

**Figure supplement 2.** Features of the tubulin binding site (see also *Supplementary file 1*).

**Figure supplement 3.** The tubulin β2-Asp and the P-loop Walker B interact with waters that occupy equivalent sites around the catalytic $Mg^{2+}$ cation.

sheet core, whereas in P-loops it points 'outside', *i.e.*, away from the protein interior – an approximately 180-degree rotation compared to Rossmanns.

The above difference in orientation relates to differences in the interactions that Rossmanns and P-loops make with parts of the ribonucleotide ligands other than their phosphate moieties. In the canonical Rossmann binding site, both the phosphate moiety and the ribose moiety are bound. The phosphate interacts with the Rossmann PBL at the N-terminus of α1 while the ribose moiety is held in place by an Asp/Glu residue at the tip of β2 (*Figure 2B*). This acidic residue forms a unique bidentate interaction with the 2' and 3' hydroxyls of the ribose moiety, and was shown to be present as Asp in the earliest Rossmann ancestor (hereafter β2-Asp) (*Laurino et al., 2016*). In P-loops, on the other hand, the core of the αβα domain does not interact with the ribose, instead making more extensive, catalytically-oriented interactions with the phosphate moieties (via the Walker A PBL, *Figure 2A*, as well as other key residues). Foremost, phospho-ligand binding also involves coordination of a metal cation, mostly $Mg^{2+}$, but also $Ca^{2+}$, by two key conserved residues: the hydroxyl of the canonical serine or threonine of the Walker A motif (GxxGxGK[**S/T**]) and an Asp/Glu residing on the tip of an adjacent β-strand. This Asp/Glu residue is the crux of the so-called 'Walker B' motif, which is typically located at the tip of either β3 or β4 (see *Shalaeva et al., 2018* for a detailed analysis).

## A shared β-(phosphate binding loop)-α evolutionary seed?

Individually, any one of the shared features described above may relate to convergence. The αβα sandwich fold has likely emerged multiple times, independently (*Medvedev et al., 2019*). The key shared functional features – namely, a phosphate binding site at the N-terminus of an α-helix and a Gly-rich phosphate binding motif – were likely favored early in protein evolution because they effectively comprise the only mode of phosphate binding that can be realized with short and simple peptides (*Longo et al., 2020*). Thus, that Rossmanns and P-loops share Gly-rich loops and the same mode of phosphate biding may also be the outcome of convergence, especially because the overall mode of binding of their phospho-ligands fundamentally differs (*Figure 2A,B*).

Curiously, however, the phosphate binding site is located in the first β-loop-α element of both Rossmanns and P-loops. In fact, the β1-α1 location of the PBL is seen not only in P-loop and Rossmann proteins, but also in Rossmann-like protein classes such as flavodoxin and HUP. However, a closer examination reveals that, although rare, phosphate binding in αβα sandwich folds can occur at alternative locations, suggesting that there is no inherent, physical constraint on its location. An illustrative example can be found in the HUP lineage (ECOD X-group 2005) – a monophyletic group of 3-layer αβα sandwich, Rossmann-like proteins that includes Class I aminoacyl tRNA synthetases (*Aravind et al., 2002b*). Most families within this lineage achieve phosphate binding at the tip of α1, as do Rossmann and P-loop proteins. However, two families, the universal stress protein (Usp) family (F-group 2005.1.1.145) and electron transport flavoprotein (ETF; F-group 2005.1.1.132) both use the tip of α4 (*Figure 3*). Intriguingly, α4, resides on the other side of the β-sheet, just opposite to α1. Accordingly, this change in the PBL's location results in a flip of ATP's phosphate groups, while preserving all other features of the binding site, including the adenine's location and the anchoring of the ribose moiety to β1 and β4 (*Figure 3* mid panel). Thus, from a purely biophysical point of view, α1 and α4 are equivalent locations for phosphate binding. Nonetheless, the ancestral phosphate binding site in both P-loops and Rossmanns resides at the tip of α1 (as judged by α4 being a rare exception). This suggests that the positioning of the PBL at the tip of α1 in both Rossmann and P-loop proteins is a signal of shared ancestry rather than convergence. As a minimum, the identification of α4 as a feasible alternative supports a model of emergence of both lineages from a seed β−PBL−αβ fragment, as outlined further below. By this scenario, α4 only emerged at a later stage, well after phosphate binding had been established at α1.

## A shared β2-Asp motif?

As outlined above, the β1-PBL-α1 segment of P-loops and Rossmanns likely represents a primordial polypeptide that could later be extended to give the modern αβα sandwich domains (*Alva et al., 2015*; *Zheng et al., 2016*). However, there are several indications that the ancestral, seeding peptide(s) of both P-loops and Rossmanns also contained β2 (*Alva et al., 2015*). In the case of the Rossmann, β2 of the seeding primordial peptide plays a functional role: an Asp at its tip forms a bidentate interaction with the hydroxyls of the nucleotide's ribose (*Figure 2B*; *Laurino et al., 2016*). The putative Rossmann common ancestor thus comprises a β-PBL-α-β-Asp fragment. Might such a fragment also be the P-loop ancestor?

In fact, both families make use of an Asp residue at the tip of the β-strand just next to β1 – in P-loops this residue is the above-mentioned Walker B motif (*Figure 2A*). Is this feature also a sought-after signature of shared ancestry? In the simplest P-loop topology, the Walker B-Asp resides at the tip of the β-strand that is adjacent to β1, as it is in Rossmanns. Thus, putting aside the connectivity of the strands, both P-loop and Rossmann possess a functional core of two adjacent strands, one from which the PBL extends and the other with an Asp at its tip (*Figure 2A,B*). However, because in P-loops the strand topology is swapped, the Walker B-Asp resides at the tip of β3 in the primary sequence (in the simplest topology described in *Figure 1C*, and at the tip of β4 in another common topology as detailed below). As elaborated later, variations in the topology of P-loops support a model by which additional β-loop-α elements got inserted into the ancestral β-PBL-α-β-Asp seed fragment such that what was initially β2 became β3 (or even β4 in other P-loop families).

However, even if we put the question of topology aside for the moment, there remains the fundamental functional difference between the P-loop Walker B-Asp (binding of a catalytic dication) and the Rossmann β2-Asp (ribose binding; *Figure 2A and B*, respectively). Can this difference be reconciled? This question might be answered by identifying cases of parallel evolution, or 'bridging

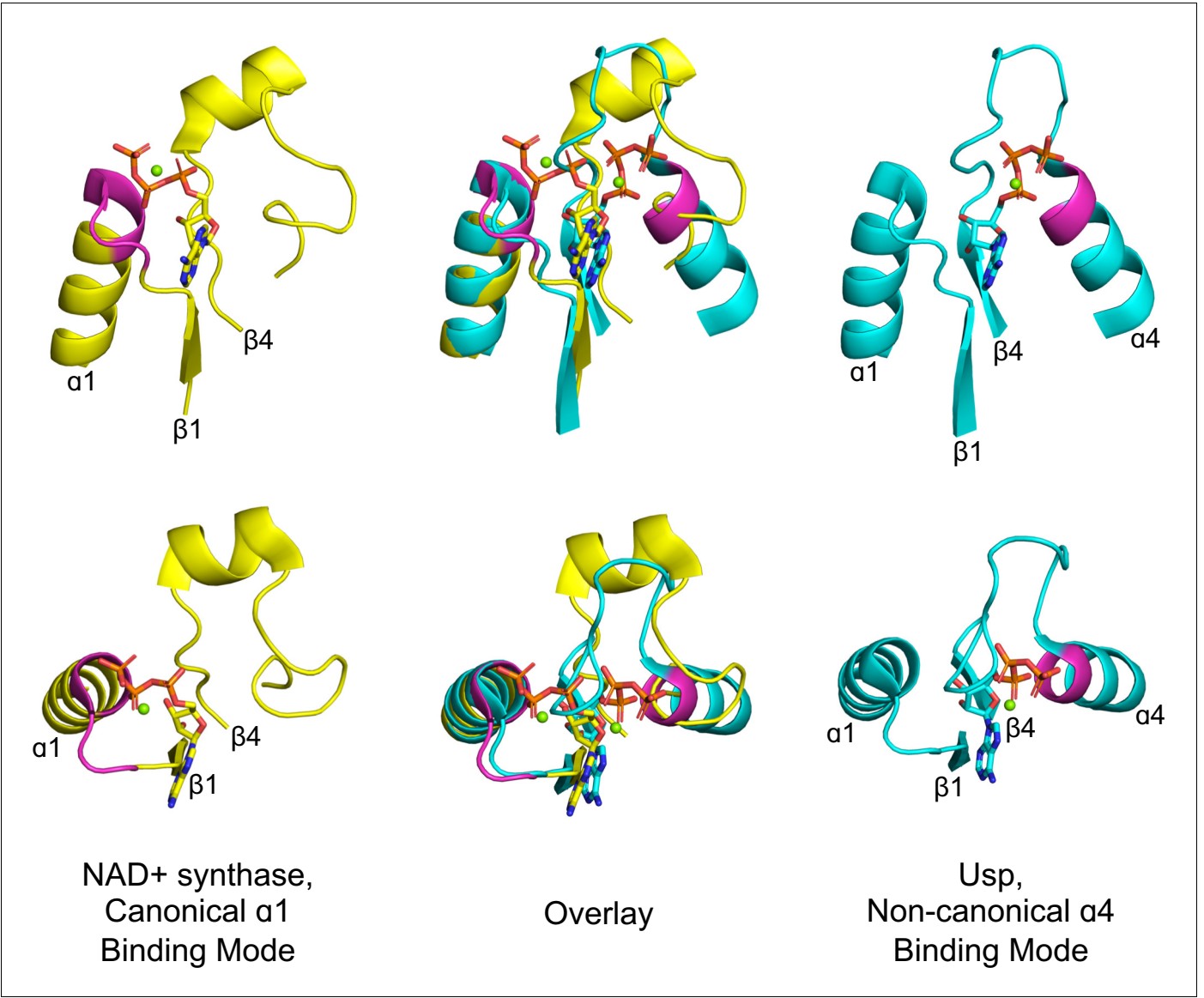

**Figure 3.** Alternative phosphate binding sites in αβα sandwich enzymes. HUP proteins are αβα sandwich proteins with a Rossmann-like strand topology. The canonical HUP phosphate binding loop is located at the tip of α1 and colored magenta (left panels; NAD$^+$ synthase; ECOD F-group 2005.1.1.13; shown is domain e1xngA1) as in Rossmann and P-loop NTPases (*Figure 2*). However, Usp (universal stress proteins) is a HUP family that exhibits kinase activity wherein phosphate binding migrated to the tip of α4 (right panels; ECOD F-group 2005.1.1.145; shown is domain e2z08A1; residues interacting with phosphate groups are colored magenta). As shown in the overlay (middle panels), despite the variation in the phosphate binding site, the ribose and adenine binding modes are identical.

proteins'. Specifically, we searched for examples of Rossmanns acting as NTPases, and examined whether they use a catalytic metal and, if so, whether this metal cation is bound by the β2-Asp.

## Tubulin – a parallelly evolved Rossmann NTPase

As explained above, Rossmanns typically use the ligand's phosphate moiety as a binding handle, whereas P-loops perform chemistry on the ligands' phosphate groups. Thus, to discover bridging proteins, we looked at the minority of Rossmann families that do act as NTPases. In all but one of these, the NTP is bound in the canonical Rossmann mode, namely with the NTP's ribose moiety bound to the β2-Asp (*Figure 2—figure supplement 1*). However, one family, tubulin, is an outlier. Tubulin is a GTPase first discovered in eukaryotes. With time, bacterial and archaeal tubulins were discovered, indicating that this lineage originated in the LUCA (*Yutin and Koonin, 2012*;

*Margolin et al., 1996*). Tubulin has undisputable hallmarks of a Rossmann (*Nogales et al., 1998a*), as noted originally (*Nogales et al., 1998b*), and is categorized as such (ECOD family: 2003.1.6.1). The strand topology is distinctly Rossmann (3-2-1-4-5), with a phosphate binding loop located between β1 and α1. We further note that binding of GTP's phosphate groups is mediated by a water molecule bound to the N-terminus of α1 (*Figure 2C*), as in canonical Rossmanns (*Bottoms et al., 2002*; *Figure 2B*) and in contrast to P-loops. However, as noticed by those who solved the first tubulin structures, GTP is oriented differently compared to the nucleotide cofactors bound by other Rossmanns (*Nogales et al., 1998a*). Our examination reveals that tubulin binds GTP in the P-loop NTPase mode – namely, with the nucleoside pointing away from the domain's core (*Figure 2C*). Indeed, tubulin's phosphate binding loop is truncated relative to other Rossmanns and adopts a conformation akin to a tight hairpin (*Figure 2*, *Figure 2—figure supplement 2A*). In fact, tubulin has a second phosphate binding loop that resides at the tip of α4 and has a critical role in catalysis, indicating that α4 can readily take the role of phosphate binding as seen in the HUP families described above (*Figure 3*).

Foremost, the β2-Asp interaction with the ribose, a hallmark of Rossmanns, is absent in tubulin (*Figure 2C*). Rather, the canonical Asp at the tip of β2 ligates a catalytic dication (*Figure 2C*). Further, tubulin's binding of the dication adopts a P-loop-like octahedral geometry (*Kanade et al., 2020*), with both the β2-Asp of tubulin and the Walker B-Asp of P-loops interacting with water molecules that occupy equivalent coordination sites (*Figure 2—figure supplement 3*). The β2-Asp is essential for tubulin's catalytic activity (*Farr and Sternlicht, 1992*), and its Walker B-like mode of action is seen across multiple tubulin structures (in many tubulins Asn is seen at the β2 tip position, though this β2-Asn also ligates the dication, either directly or via a water molecule [*Figure 2—figure supplement 2B*; *Supplementary file 1*]).

Tubulin therefore comprises an intriguing case of a Rossmann that evolved an NTPase function by reorienting the NTP substrate to bind in the P-loop NTPases mode and repurposing the canonical Rossmann β2-Asp to ligate a catalytic metal cation. Put differently, tubulin shows that the functional differences between the P-loop Walker B and the Rossmann β2-Asp can be reconciled.

## Shared *theme*s between Rossmanns and P-loops

Encouraged by tubulin, we endeavored to look for additional evidence for bridging proteins, ideally with respect to not only structure but also sequence. To this end, we employed the concept of a *bridging theme* – short stretches for which alignments are statistically significant (≥20 residues; HHSearch E-value <$10^{-3}$) yet with the flanking regions showing no detectable sequence homology (*Kolodny et al., 2020*). In the context of this work, we specifically searched for shared themes in structures that belong to Rossmanns (X-Group 2003) on the one hand and P-loops (X-group 2004) on the other. By focusing the sequence homology search on evolutionarily-distinct domains, and by using bait sequences derived from validated sequence themes (*Nepomnyachiy et al., 2017*), the sensitivity and accuracy of this approach exceeds that of standard HMM-based searches (further details about themes detected between other X-groups are described in a forthcoming manuscript *Kolodny et al., 2020*). Given a stringent statistical threshold, only a few shared themes were detected, all involving the P-loop enzyme HPr kinase/phosphatase (F-Group 2004.1.2.1; PDB: 1ko7; *Figure 4A*). A few different Rossmann F-groups share a theme with this P-loop, with sorbitol dehydrogenase (F-Group 2003.1.1.417; PDB: 1k2w) and short chain dehydrogenase (F-Group 2003.1.1.332; PDB: 3tjr) showing the greatest overlap (*Figure 4*).

HPr kinase/phosphatase is a bifunctional bacterial enzyme that catalyzes the phosphorylation and dephosphorylation of a signaling protein (HPr; *Márquez et al., 2002*). The P-loop domain comprises its C-terminal domain and carries the kinase function (hereafter Hpr kinase). Remarkably, the Walker B-Asp of Hpr kinase resides at the tip of β2, rather than at β3 or β4 as in the canonical P-loops. Consequently, although no such constraint or steering was applied to the search algorithm, the detected shared theme encompasses an intact β1-PBL-α1-β2-Asp element in the Rossmann proteins (where this element is canonical) as well as in this unique P-loop family (*Figure 4A*). As expected, this element is conserved in both the P-loop Hpr kinase and in the Rossmann families, with the Gly residues of the phosphate binding motifs, and the β2-Asp's being almost entirely conserved (*Figure 4B*). This result underscores the significance of the β1-PBL-α1-β2-Asp motif as the shared evolutionary seed of both Rossmanns and P-loops (detailed in the next section).

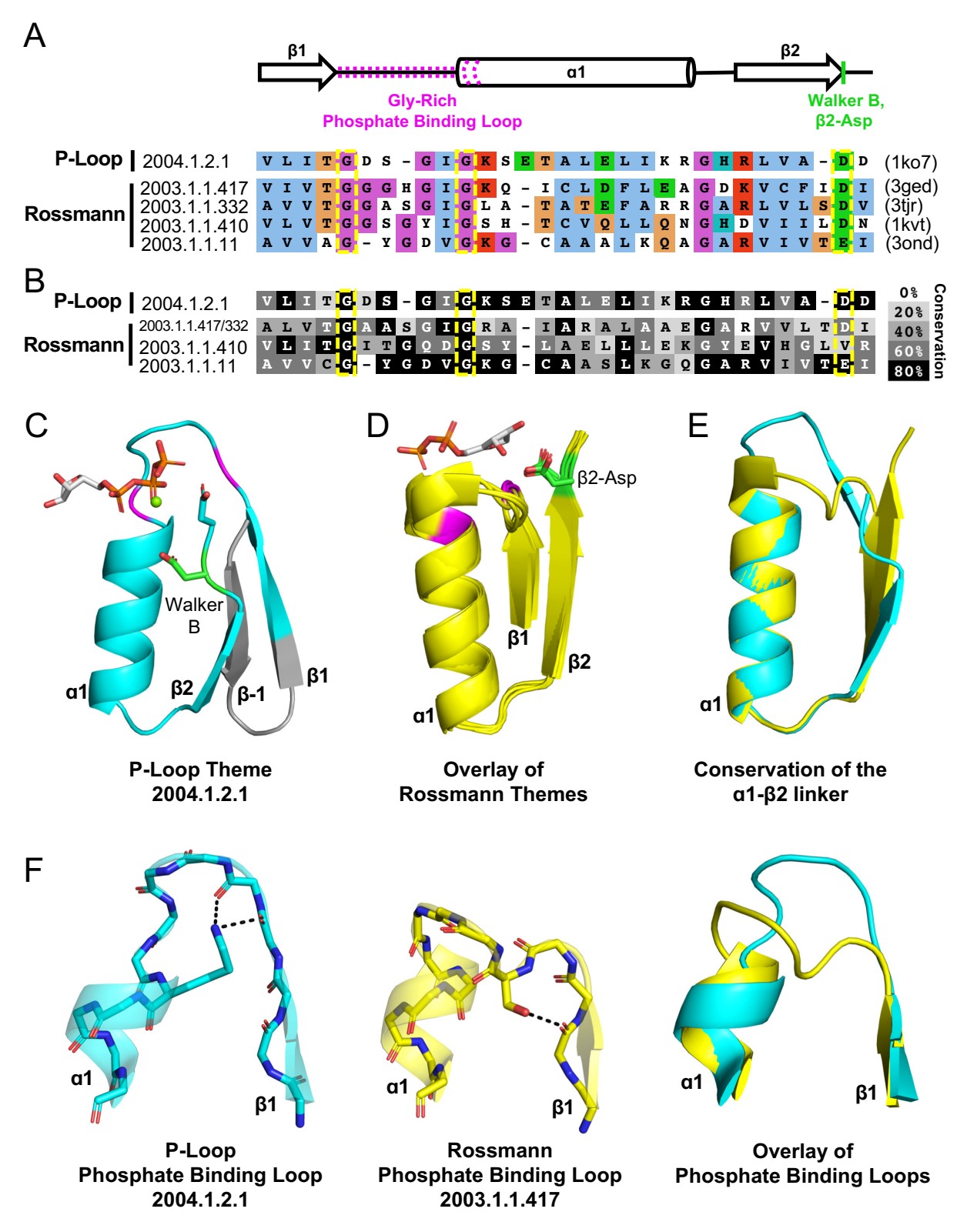

**Figure 4.** Theme sharing between Rossmann and P-loop enzymes. (**A**) Sequence alignment of the shared themes. PDB codes are shown on the right, and the ECOD F-group to which they belong are on the left. The identified themes involve a segment of a single P-loop NTPase, Hpr kinase (top line, ECOD domain e1ko7A1), that aligns to a variety of Rossmanns that belong to four different F-groups (representatives shown here; see *Supplementary file 3* for the complete list of bridging themes). (**B**) The consensus sequence of each F-group (see Methods) is shaded according to the

*Figure 4 continued on next page*

*Figure 4 continued*

degree of conservation. The individual sequences identified by the theme search show higher similarity by default, yet nonetheless, the family consensus sequences also align well, and the identical residues tend to be conserved. (C-D) Although detection of the shared theme was based on sequence only, structurally, the shared theme encompasses the β1-PBL-α1-β2-Asp element in both the P-loop protein (panel C; Hpr kinase, ECOD domain e1ko7A1) and the theme-related Rossmanns (panel D; ECOD domains e3gedA1, e1kvtA1, e3ondA1, and e3tjrA1; the ligand is bound by domain e3tjrA1). Note that only the pyrophosphate and ribose moieties of the ligand are shown for clarity. The conserved phosphate binding loop glycine residues are colored magenta and the β2-Asp is colored green. For panel C, the ligand binding mode was modeled using the structure of a liganded P-loop protein (see *Methods*). (E) An overlay of the β1-PBL-α1-β2-Asp element of the Hpr Kinase (cyan; ECOD domain e1ko7A1) and one of the theme-related Rossmann dehydrogenases (yellow; ECOD domain e3tjrA1). (F) Structural details of the phosphate binding loops: The Walker A binding loop of Hrp kinase (left panel; ECOD domain e1ko7A1); the phosphate binding loop of sorbitol dehydrogenase (middle panel; ECOD domain e1k2wA1); and an overlay of both loops (right panel).

The online version of this article includes the following figure supplement(s) for figure 4:

**Figure supplement 1.** Topological diversity in the P-loop evolutionary lineage.

Consistent with the idea of parallel evolution, these bridging P-loop and Rossmann proteins seem to be at the fringes of their respective lineages. In the case of HPr kinase, the active site is characterized by a canonical Walker A motif but the Walker B-Asp is uncharacteristically situated at the tip of β2 (*Figure 4D*). Further, in P-loop families with the simplest topology, the Walker B-Asp resides at the tip of a β-strand that structurally resides next to β1 (β3, *Figure 2A*). However, in most P-loops, another strand, typically β4, is inserted between β1 and the strand carrying the Walker B-Asp (*Figure 4—figure supplement 1*; *Leipe et al., 2003*). Hpr kinase belongs to this second category; however, its intervening strand is highly unusual – an anti-parallel β-strand inserted between β1 and β2. In the primary sequence, the intervening strand is an N-terminal extension, and thus upstream of β1 (*Figure 4C*; annotated as β−1). Indeed, HPr kinase is classified as an outlier with respect to the greater space of P-loop proteins. The P-loop X-group in ECOD (X-group 2004) is split into two topology groups (T-groups): *P-loop containing nucleoside triphosphate hydrolases*, which includes 196 F-groups that represent the abundant, canonical P-loop proteins, and *PEP carboxykinase catalytic C-terminal domain*, which is comprised of just three F-groups. HPr kinase is classified as part of the latter. As discussed below, the variation in topology of HPr also highlights the structural plasticity of the P-loop fold with respect to insertions.

The sorbitol dehydrogenase and short chain dehydrogenase both have the canonical Rossmann strand topology and β2-Asp. Homology modeling of the enzyme-NAD$^+$ complex, and inspection of closely related structures, suggest that binding of the NAD$^+$ cofactor is also canonical (*Figure 4D*). However, the PBL of these three Rossmann proteins is nonstandard (GxxxGxG instead of the canonical Rossmann which is GxGxxG; *Figure 4A*). Further, although the structural positioning of the last two glycine residues is rather similar to that of canonical Rossmann proteins, the extended GxxxGxG motif results in an extended PBL with higher resemblance to the P-loop (*Figure 4F*). Indeed, a sequence alignment reveals that Hpr's P-loop (which is canonical) and these nonstandard Rossmann PBLs are only a few mutations away from each other (*Figure 4A and F*, overlay in right panel).

## An ancestral βαβ seed of both Rossmanns and P-loops

The above findings support the notion of a common Rossmann/P-loop ancestor, the minimal structure of which is βαβ. This ancestral polypeptide includes just two functional motifs: a phosphate binding loop and an Asp, which could play a dual role of either binding the ribose moiety of various nucleotides or of ligating a dication such as Mg$^{2+}$ or Ca$^{2+}$. Previously, such a polypeptide (*i.e.*, β-PBL-α-β-Asp) has been proposed as the seed from which Rossmann enzymes emerged (*Alva et al., 2015*; *Laurino et al., 2016* and references therein). In contrast, a βα element was assigned as the P-loop ancestral seed (*i.e.*, a β-P-loop-α segment; *Alva et al., 2015*; *Laurino et al., 2016*., and references therein). Here, we argue that ancestral polypeptide(s) comprising a βαβ element gave rise to both lineages, and possibly that a single polypeptide served as a common ancestor of both lineages.

## From an ancestral seed to intact domains

We further hypothesize that the above seed fragment was subsequently expanded by addition of β-loop-α and/or α-loop-β elements. Such expansion would have enabled a functional split, or sub-

specialization, of the two separate lineages, Rossmann and P-loop, which then further evolved and massively diversified. In essence, this split regards two key elements – phospho-ligand binding and β-strand topology. Our analysis indicates the feasibility of both.

The plausibility of common descent of the P-loop Walker A and the Rossmann phosphate binding loops is indicated by the detected shared theme described above (*Figure 4*). Although the canonical motifs of both lineages differ, there still exists – particularly among Rossmann proteins – alternative motifs that could diverge via a few mutations to a Walker A P-loop. Other Rossmanns possess a GxGGxG motif that also represents a potential jumping board to a Walker A P-loop (*Zheng et al., 2016*). Given that it mediates binding rather than catalysis, and owing to its lower conservation, we speculate that a Rossmann PBL could be replaced by a Walker A-like P-loop. However, at present, how permissive the Rossmann PBL is to sequence changes that will render it Walker A-like is unclear. Future experimental work, possibly using the Rossmann enzymes indicated here (*Figure 4*), might lend support to the hypothesis that these two PBLs are indeed evolutionarily related. The identified shared themes also indicate that the β2-Asp of the presumed ancestral fragment could not only bind the ribose moiety as in Rossmanns, but also serve as a Walker B, as in P-loops. Tubulin lends further support, indicating that a β2-Asp can indeed play a dual role.

Expansion of the ancestral βαβ fragment would enable not only the sub-functionalization of the two functional elements described above, but also the fixation of two separate β-strand topologies – the sequential Rossmann topology *versus* the swapped P-loop one. Both folds are, in essence, a tandem repeat of β-loop-α elements (*Figure 1*). The evolutionary history of other repeat folds tells us that expansion typically occurs by duplication of the ancestral fragment (*Eck and Dayhoff, 1966*; *Romero Romero et al., 2018*; *Zhu et al., 2016*; *Longo, 2020*), or parts of it, but also by fusion of independently emerging fragments (*Grishin, 2001*; *Setiyaputra et al., 2011*). Regardless of the origin of the extending fragment(s), given a βαβ ancestor, similar processes could have given rise to both of these folds. As summarized in *Figure 5*, in both P-loops and Rossmanns, the first newly added β-strand would pack against the ancestral β-strand bearing the Asp residue – irrespective of whether the incoming β-strand was the result of a sequence insertion or a C-terminal extension

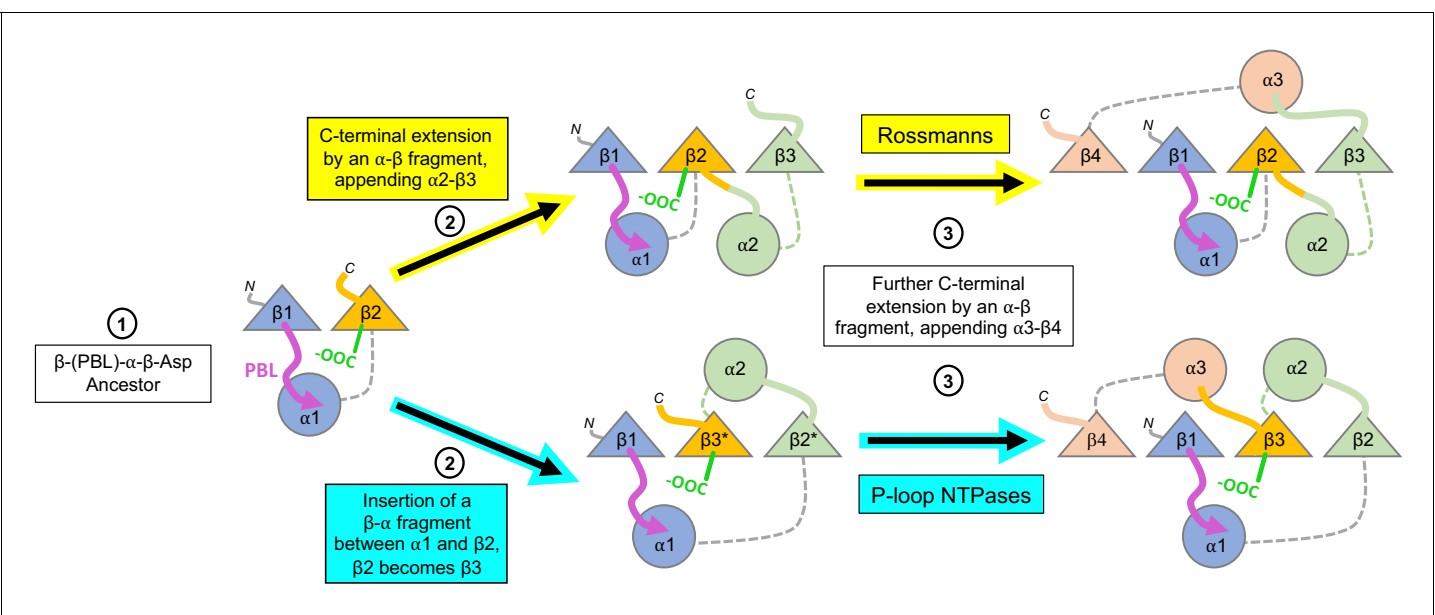

**Figure 5.** Divergence of the Rossmann and P-loop NTPase folds from a common ancestral polypeptide. Emergence begins with a presumed β-PBL-α-β-Asp ancestor that could act as either a Rossmann or a P-loop NTPase, depending on how the phospho-ligands bind and the role taken by the β2-Asp (*Figure 2A and C*). In the second step, the ancestral fragment is either extended at its C-terminus by fusion of an αβ fragment to generate a Rossmann-like domain (top row); or, by insertion of a βα fragment between α1 and β2 to yield a P-loop-like domain (bottom row). Note that βα fragment sequence insertion results in the ancestral β2 that carries the Walker B-Asp becoming β3. Note also that the location of the newly added helix, α2, differs: It can pack next to α1, as in the Rossmann fold; or, it can migrate to the opposite side of the β-sheet, as in the P-loop NTPase fold. Following *Figure 1*, the top loops are shown as thick lines while the bottom loops are shown as thin, dashed lines. The phosphate binding loop is colored magenta, and the β2-Asp/Walker B-Asp is shown in green.

(*Step 2*). In the case of a C-terminal extension by an αβ fragment, the strand bearing the β2-Asp is unperturbed and the strand topology is Rossmann-like; conversely, a sequence insertion of a βα element between the two ancestral β-strands results in a P-loop topology, with the β-strand that carries the Walker B-Asp becoming β3. In the next extension, the newly added 4th strand packs against the other side of growing domain, next to β1 (*Step 3*; β4 added with its preceding helix, α3). Subsequent strand(s), β5 and onward, could in principle be added sequentially, one next to the other, to yield the intact domains as we know them today. Indeed, evidence that extensions at the edges of the core β-sheet can happen in both folds is provided by the existence of Rossmann and P-loop proteins with either 5 or six strands. Further, circular permutations are common in Rossmanns, as are non-canonical additions of a 7th strand, including anti-parallel strands, at either end of the β-sheet (*Grishin, 2001*).

Upon establishment of the core domain, transitions in the topology of β-sheets that result from strand swaps, or strand invasions, as has been documented (*Grishin, 2001*), were likely key drivers of divergence and sub-functionalization. In particular, the P-loop lineage seems to have undergone various strand swaps and insertions that gave rise to a variety of topologies (*Leipe et al., 2003*), including the noncanonical one, with an antiparallel strand, seen in Hpr kinases (*Figure 4C*). Indeed, a survey of P-loop F-groups reveals multiple strand topologies (*Figure 4—figure supplement 1* and *Leipe et al., 2003*). In general, families that catalyze phosphoryl transfer, namely kinases such as thymidylate kinase (F-group 2004.1.1.166), but also GTPases such as elongation factor Tu (F-group 2004.1.1.258), tend to have the simplest 2-3-1-4-5 topology illustrated in *Figure 1C* (see *Leipe et al., 2003*). In these proteins, the Walker A P-loop and the Walker B-Asp reside on adjacent strands, with the Walker B motif on the tip of β3 (as illustrated in *Figure 2A*). On the other hand, 'motor proteins', in which ATPase activity drives a large conformational change that turns into some further action, such as helicases or the ATP cassette of ABC transporters, tend to have a strand inserted between β1 and β3, to yield a 2-3-4-1-5 topology. Here, the Walker B-Asp is also situated on the tip of β3, yet with an intervening strand (β4) between the Walker A and Walker B motifs. The split between these topologies is ancient, likely predating the LUCA (*Leipe et al., 2003*), and supports the hypothesis that early events of fusions as well as insertions were associated with the functional radiation of the P-loop lineage.

In contrast to the P-loop's variable topologies, the pseudo-symmetrical Rossmann topology seems highly conserved (in a previous analysis of the Rossmann fold, we did not detect a single structure annotated as Rossmann with a swapped strand topology *Laurino et al., 2016*). Further, the very same topology appears in other domains, so-called Rossmann-like, or Rossmannoid domains (foremost, flavodoxin, 2-1-3-4-5, and HUP, 3-2-1-4-5). The latter two also represent ancient, pre-LUCA phospho-ligand binding domains that likely evolved independently of the Rossmann (*Medvedev et al., 2019*) and converged to the same topology. That convergence to the Rossmann topology occurred frequently may relate to the higher thermodynamic and/or kinetic stability of the symmetric strand topology. Indeed, the design of Rossmann-like proteins is readily realized compared to P-loops-like proteins with the swapped strand topology (*Romero Romero et al., 2018*). Furthermore, systematic assays of refoldability of the *E. coli* proteome, and a comparison of the folds to which these proteins belong, indicated that Rossmann is among the most refoldable folds while P-loop NTPases are among the poorly refolding ones (*To et al., 2020*).

## Concluding remarks

Protein evolution spans nearly 4 billion years, with the founding events occurring pre-LUCA. As such, for many protein families, definitive assignment of homologous versus analogous relationships (shared ancestry versus convergent evolution) may never be possible (*Aravind et al., 2002a*). Confounding matters further, early constraints on protein sequence and structure have further limited the number of possible solutions to a subset of structures and binding motifs (*Longo et al., 2020*), making convergence a more likely scenario, particularly in the most ancient proteins. Thus, although discovered several decades ago, whether the P-Loop NTPase and Rossmann lineages diverged or converged remains an open question. The availability of thousands of structures, highly curated databases that catalogue them (*Chandonia et al., 2017*; *Cheng et al., 2014*), and sensitive search methods (*Hancock et al., 2004*) and algorithms (*Nepomnyachiy et al., 2014*) allows this question to be reexamined. Here, evidence in favor of common ancestry between these lineages is provided, though convergence cannot and should not be entirely ruled out. Whether it was convergence or

divergence, our analysis suggests that both lineages emerged from a polypeptide comprising a β-PBL-α-β-Asp fragment. Such a polypeptide was likely the ancestor of both P-loops and Rossmanns – be it the same polypeptide, or two (or more) independently emerged ones. Reconstruction of ancestral polypeptides (*Longo, 2020*), including 40 residue polypeptides that relate to the P-loop NTPase ancestor (*Vyas, 2020*), may allow us to further examine the common *versus* independent emergence scenarios.

## Materials and methods

### The functional diversity of the P-loop and Rossmann lineages

In total, three X-groups comprising 663 ECOD F-groups were analyzed (*Supplementary file 2*): P-loop-like (X-group 2004; 157 F-groups), Rossmann-like (X-group 2003; 168 F-groups) and Rossmann-like structures with the crossover (X-group 2111; 338 F-groups). For this analysis, ECOD version develop210 was used and X-groups 2003 and 2111 were merged. The sequences of each F-group (70% identity cutoff) were mapped to a SUPERFAMILY (*Wilson et al., 2009*) entry with HMMsearch (*Hancock et al., 2004*) using the HMM profiles provided by the SUPERFAMILY database. The SUPERFAMILY EC2Domain mapping file was used to collect the Enzyme Commission (EC) classes associated with each family. In total, we identified 75 EC classes associated with P-loops (X-group 2004) and 727 with Rossmanns (X-groups 2003 and 2111). Within all three X-groups, the majority of families exhibit transferase activity (2.-.-.-). Within the Rossmann-like X-group, oxidoreductases (1.-.-.-) are also common. For both P-loop and Rossmann-like structures with the crossover, the second most common enzyme activity is hydrolase (3.-.-.-), while for Rossmann-like families, hydrolase activity is the least common.

### Identification of shared themes between P-loops and Rossmanns

We used HHSearch (version 3.0.0) (*Hildebrand et al., 2009*) to compare a set of previously curated themes (*Nepomnyachiy et al., 2017*) to a 70% non-redundant set of ECOD domains (version develop210). Using an E-value threshold of $10^{-3}$, a coverage threshold of 85% (for the local alignment), and a minimal length of 20 residues, we identified 267 themes with significant hits to proteins belonging to both ECOD X-groups 2004.1 (P-loop domains-like) and 2003.1 (Rossmann-like). All of these themes matched the same P-loop domain (e1ko7A1) with various Rossmann domains. To reduce the extensive redundancy among the themes, which in turn leads to redundancy in the detected proteins, we kept only two representatives Rossmanns per theme. To identify the representative domains, we re-aligned the parts matching the theme using a Smith-Waterman (SW) or Needleman-Wunsch (NW) alignment, and the parts before and after the theme using an SW alignment. The representative domains are the ones with the most similar matching parts and the most dissimilar flanking parts. A p-value for the aligned parts was calculated from the significance of the alignment score relative to scores from alignments of random segments. Here, we estimated the parameters of the extreme value distribution (EVD) from the scores of the alignments between one of the two well-aligned segments and 1000 randomly chosen segments drawn from a multinomial distribution estimated from the other of the two well-aligned segments. We kept only cases where the matching parts have a score with a p-value lower than 0.05. This procedure resulted in a set of 57 Rossmann domains, each aligned to the Hpr kinase (PDB: 1ko7). For 50 of these 57 hits, the matching parts were aligned with the SW local alignment and the rest were aligned by NW global alignment. Here, we report the 51 cases that match the β1-α1-β2 regions (26 from the ECOD family 2003.1.1.417, 18 from 2003.1.1.332, 5 from 2003.1.1.410, and 2 from 2003.1.1.11; *Supplementary file 3*) The alignments presented in the manuscript are the global alignments recalculated for the β-PBL-α-β elements in themes that bridge the two evolutionary lineages.

### Modeling ligand placements in unliganded structures

HrP kinase (e1ko7A1) does not have a ligand bound. The conformation of the PBL, however, is canonical. Thus, despite no structure from the same F-group having a relevant ligand bound, the overall positioning of the ligand can be estimated by overlaying a canonical PBL with a bound ligand from a different P-loop F-group. To generate *Figure 4C*, the PBL and ligand from ECOD domain e6at2A2, corresponding to residues 247–257, was aligned to residues 150–160 in chain A of 1ko7.

This structure was chosen because it is in the same T-group as HrP kinase (2004.1.2.-) and, although the two domains have nearly undetectable sequence identity, they share the same general topology, including the inserted anti-parallel β-strand adjacent to β1. The sequences of the PBLs (F<u>G</u>LS<u>GT</u>G<u>K</u>T<u>T</u>L and V<u>G</u>PN<u>GS</u>G<u>K</u>S<u>T</u>V for 6at2 and 1ko7, respectively; identical residues underlined) show high similarity as do the structures of the PBLs that were aligned to generate the modeled ligand (Cα RMSD of 0.49 Å).

## Calculating consensus sequences and residue conservation scores

The relevant ECOD F-groups (*Figure 4*) were mapped to the corresponding Pfam families. Since 2003.1.1.417 and 2003.1.1.332 are associated with one Pfam family, they were analyzed jointly. Seed alignments were extracted from Pfam, clustered at 70% sequence identity using CD-HIT (*Fu et al., 2012*), and the consensus sequence and conservation scores were calculated for the shared region (theme) using JalView.

# Acknowledgements

This research has been supported by Grant 94747 by the Volkswagen Foundation. NB-T's research is supported in part by the Abraham E Kazan Chair in Structural Biology, Tel Aviv University. DST is the Nella and Leon Benoziyo Professor of Biochemistry. We are grateful to Ita Gruic-Sovulj for her role in the analysis of the HUP domain that led to *Figure 3*, and to Andrei Lupas for insightful and critical comments.

# Additional information

## Funding

| Funder | Grant reference number | Author |
|---|---|---|
| Volkswagen Foundation | 94747 | Dan S Tawfik<br>Rachel Kolodny<br>Nir Ben-Tal |

The funders had no role in study design, data collection and interpretation, or the decision to submit the work for publication.

## Author contributions

Liam M Longo, Dan S Tawfik, Conceptualization, Formal analysis, Visualization, Writing - original draft, Writing - review and editing; Jagoda Jabłońska, Manil Kanade, Formal analysis, Visualization; Pratik Vyas, Conceptualization; Rachel Kolodny, Conceptualization, Formal analysis, Visualization, Methodology, Writing - review and editing; Nir Ben-Tal, Conceptualization, Formal analysis, Methodology, Writing - review and editing

## Author ORCIDs

Liam M Longo https://orcid.org/0000-0002-1773-0611
Jagoda Jabłońska https://orcid.org/0000-0001-5455-7451
Pratik Vyas https://orcid.org/0000-0002-8961-5575
Manil Kanade http://orcid.org/0000-0002-0076-8584
Rachel Kolodny https://orcid.org/0000-0001-8523-1614
Nir Ben-Tal https://orcid.org/0000-0001-6901-832X
Dan S Tawfik https://orcid.org/0000-0002-5914-8240

## Decision letter and Author response

Decision letter https://doi.org/10.7554/eLife.64415.sa1
Author response https://doi.org/10.7554/eLife.64415.sa2

## Additional files

### Supplementary files

- Supplementary file 1. Dication binding in tubulins.
- Supplementary file 2. Supplementary file 2.
- Supplementary file 3. Summary of Rossmann proteins that share a bridging theme with the P-Loop domain e1ko7A1.
- Transparent reporting form

### Data availability

All data analysed in this manuscript are from publicly available archives such as the Protein Databank.

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
