## [Decision Letter]

[Editors' note: this paper was reviewed by Review Commons.]

**Acceptance summary:**

The paper is an elegant and well written study looking at whether the P-loop NTPases and the Rossman superfamilies are evolutionarily related and gives strong evidence that this may indeed be the case. I was particularly struck by the even-handed way the authors describe their work including explaining that despite all the evidence they present for divergence they cannot rule out convergent evolution. This paper not only describes the potential evolutionary relationship of two major superfamilies for the first time but should also help others who want to look for evolutionary relationships between very distantly related protein superfamilies.

---

## [Author Response]

Reviewer #1 (Evidence, reproducibility and clarity (Required)):This is a fascinating and beautifully written article about the possible evolutionary relationship between two major protein superfamilies – the P-loop NTPases and the Rossmans. Both are ancient and highly diverse superfamilies, containing a significant proportion of all extant domain sequences and were probably amongst the earliest enzyme superfamilies to emerge in evolution. No major evolutionary classification of proteins, such as SCOP, reports evolutionary relationships between them.Both share the same structural architecture of a β-α-β 3-layer sandwich and have an intriguing number of other shared structural features including the location of the binding site for phospho-ligands. However, whilst both bind phosphorylated ribonucleosides, the mode of binding differs and also the manner in which these compounds are exploited. Furthermore, there are differences in the topologies of the folds possibly suggesting distinct evolutionary trajectories. The Rossmanns appear to be more structurally conserved, whilst the P-Loops vary more in their topologies and possibly represent less stable arrangements of β-sheets and α-helices.The authors have brought together several strands of evidence to explore possibly evolutionary relationships. Detailed structural analyses allow the authors to explicitly detail the significant shared structural features. For example, similarities in the mode of binding the phosphate moiety in the ligand. The structural features are well described and there are appropriate illustrations visualising key differences and similarities.The shared features of the phosphate binding site likely emerged and were favoured early in evolution, as supported by other analyses reported by Longo et al. However, as the authors point out there are other compelling similarities including the equivalent location of this site in the first β-loop-α element in both superfamilies, which is not a necessary constraint of phosphate binding and the authors support this by giving examples of phosphate binding at the tip of α-4. In addition, they provide evidence supporting the common involvement of β-2 which contains the conserved Asp in the Rossmanns common ancestor. The Walker-B Asp in the P-loops is also at the tip of the β-strand adjacent to β-1, as in the Rossmanns – although this is an inserted strand relative to the Rossmann topology. The authors propose feasible evolutionary scenarios for how the P-Loops and Rossmans may have diverged to acquire additional secondary structure elements extending the common β-PBL-α-β-Asp feature present in both superfamilies.Further compelling evidence is given by detection of a bridging protein – Tubulin – linking the two superfamilies. This has the distinct Rossmann topology but binds GTP in the P-loop NTPase mode. Furthermore, the GTP is hydrolysed by water activated by a ligated metal dication. Final support is given by reporting common sequence themes between the P-loop enzyme HPr kinase/phosphatase and some Rossmann proteins. The authors present further interesting and detailed analyses of similarities between the proteins sharing this unusual theme.The evidence provided by the authors for the shared β-PBL-α-β-Asp fragment seems very strong to me and has been presented in an interesting and informative way. Of course, it is not possible to know the subsequent evolutionary trajectories but the scenarios presented seem plausible.

We thank the reviewer for their encouraging remarks on our manuscript.

I only have minor comments1) SCOP2 provides information on links between superfamilies based on rare sequence or structural features. Have the authors checked this resource for any details on β-PBL-α-β-ASP fragment? Or perhaps consulted with Alexey Murzin about this feature?

The classification of Rossmann and P-Loop proteins in SCOP2 is consistent with the ECOD classification scheme. For further confirmation, we wrote Alexey Murzin and he replied that Rosmanns and P-Loops are annotated as two separate evolutionary lineages, termed “hyperfamilies” in SCOP2. He found our new evidence compelling, but that given the current criteria for shared ancestry, P-loops and Rossmanns are separate lineages.

2) I was rather confused by the way in which EC annotations were collected for the two superfamilies ie via Pfam – wouldn’t it be better to use SUPERFAMILY as the domain structures would map directly to these sequence relatives. I’m also surprised that they only took the common EC from a Pfam family since the aim of this analysis was to identify how many different enzyme functions the two superfamilies supported. Pfam does not classify by function and so inevitably groups functionally diverse relatives. However, to get the full range of enzyme functions supported by these superfamilies I would have thought all non-redundant EC functions across these constituent Pfam families should be counted. Perhaps I have misunderstood.

We have updated the analysis to make use of the SUPERFAMILY database and, as per your suggestion, we now count all non-redundant EC numbers. Although the EC number counts have somewhat changed, the major point – that these are exceptionally diverse evolutionary lineages – has not.

3) The authors refer to a set of previously curated “themes” and allude to a methodology that will be reported in a forthcoming manuscript. The idea of identifying rare themes and then using them to locate very distant homologues is appealing. However, I think some details should be provided here. For example, some brief details on the technology for detecting the themes and thresholds on significance. How rare are they and how conserved do these fragments need to be between superfamilies to join their curated list? Furthermore, how many of these curated themes are similar to the one reported in their article and do they get crosslinks to other superfamilies based on closely related themes? ie how unique is this theme to the P-loop and Rossmanns and are there closely related themes linking these two superfamilies to other superfamilies? I would imagine it is quite a distinct theme but I would have liked to see a few more details on this to reassure that there are no closely related themes.

We have updated the manuscript to include a more detailed description of the methods used to detect bridging themes shared between the Rossmann and P-Loop evolutionary lineages. In addition, we now include a supplemental table (Supplementary file 2) with all of the initial hits from the theme analysis.

4) The authors have built model structures to allow them to estimate ligand location in proteins with no structural characterisation. It would be helpful if they reported the degree of sequence similarity between the query and template proteins and also the model quality.

We have updated this section to include more details. In addition, we have identified a structure from the same T-group to serve as our ligand donor. The updated ligand donor is more closely related to 1ko7 than the previous ligand donor, though the positioning of the ligand is effectively unchanged. We note that the global sequence identity to both the previous and new ligand donor is low (less than 30% sequence identity). However, the phosphate binding loops align well in both sequence and structure, as is detailed in the revised Materials and methods section.

Reviewer #3 (Evidence, reproducibility and clarity (Required)):The study by Longo et al. was devoted to evolutionary history of P-loop NTPases and Rossmann fold proteins. Although not related in sequence, the two protein families share some structural features that imply that they could be diverged from a common ancestor. Using bioinformatic analyses, the study under review identified some bridge proteins (of tubulin family) that share themes of both P-loops and Rossmanns, offering a possible support for the common ancestry. A minimum ancestral peptide structure is proposed based on the analysis and its possible diversification trajectory is hypothesized.Even though the divergence scenario is clearly outlined, the authors do not over-interpret the observations and admit that convergence could still explain the scenario. The methodology and results are sufficiently described and conclusions are explained in detail. Although it would be really interesting to design an experimental study to support the conclusion (and I suppose that the authors will do that), that is clearly outside the scope of this bioinformatic study.

Obtaining experimental evidence for our hypothesis is far from trivial. Modern proteins, including the bridging ones identified here, may not be amenable to exchange due to differing contexts (epistasis). Still, we agree that highlighting experimental directions is a good idea. We have updated the sections *From an ancestral seed to intact domains* and *Conclusion* to include a brief discussion of experiments that may help test our hypotheses about the evolution of these protein lineages.

I would not propose any major changes to the manuscript as I think that the message is very clear.Minor comments:1) In the Results section, the text is very clear but tends to be repetitive in places. I think the manuscript would be more easily readable if more to the point at some sections.

We have edited the manuscript to remove cases of unnecessary repetition in the Results section and throughout.

2) There is probably a few typos or unclear sentences, e.g. "The core, most common topology…); three lines from the bottom "(where this element in canonical", probably should be "is canonical"; "the mode of binding of the catalytic dication of tubuling (often Ca^2+^)" – all the structures listed in Supplementary file 1 list Mg^2+^, so "often" is a bit misleading.

We have corrected the unclear sentences and typos noted above, as well as a few others.